# Design and Analysis of Broadband LiNbO_3_ Optical Waveguide Electric Field Sensor with Tapered Antenna

**DOI:** 10.3390/s21113672

**Published:** 2021-05-25

**Authors:** Haiying Lu, Yingna Li, Jiahong Zhang

**Affiliations:** 1Faculty of Information Engineering and Automation, Kunming University of Science and Technology, Jingming South Road No.727, Chenggong District, Kunming 650500, China; haiyinglu@stu.kust.edu.cn (H.L.); liyingna@kust.edu.cn (Y.L.); 2Yunnan Key Laboratory of Computer Technology Applications, Kunming 650500, China

**Keywords:** integrated optics, electric field sensor, resonant frequency, LiNbO_3_ substrate, relative permittivity

## Abstract

The three-dimensional (3D) simulation model of a lithium niobate (LiNbO_3_, LN) optical waveguide (OWG) electric field sensor has been established by using the full-wave electromagnetic simulation software. The influences of the LN substrate and the packaging material on the resonance frequency of the integrated OWG electric field sensor have been simulated and analyzed. The simulation results show that the thickness of the LN substrate has a great influence on the resonant frequency of the sensor (≈33.4%). A sensor with a substrate thickness of 1 mm has been designed, fabricated, and experimentally investigated. Experimental results indicate that the measured resonance frequency is 7.5 GHz, which nearly coincides with the simulation results. Moreover, the sensor can be used for the measurement of the nanosecond electromagnetic impulse (NEMP) in the time domain from 1.29 kV/m to 100.97 kV/m.

## 1. Introduction

With the rapid development of the electric power industry, the electromagnetic (EM) field has a great impact on our lives and the development of society. The influence of the EM field has aroused extensive attention, especially in the area of communication, aerospace, national defense, electric power [1,2,3,4]. An intense pulsed electric field may cause temporary failures or permanent damage to electronic equipment and systems, such as radar, navigation and computer [5,6,7]. Therefore, the measurement of an intense pulsed electric field is becoming more and more important.

In the 1978s, the D-dot sensor was designed to measure the transient EM pulse [8]. However, as some independent metal components, such as an integrator and a balancer, are required, and the electrical cable is used for the signal transmission, the EM interference cannot be ignored, especially the coupled interference. The active electro-optic (EO) modulation type sensor has been developed, and optical fiber is used for signal transmission to reduce EM interference [9,10]. Generally, the active EO modulation type sensor receives an electric field signal through an antenna. Thus, by designing the antenna structure, the sensitivity of the sensor can be effectively improved [11,12]. However, the disturbance to the original electric field that comes from the metallic antenna structure still exists and cannot be ignored [13].

The optical electric field sensors based on the Pockels or Kerr effect can overcome these shortcomings effectively because of the all-dielectric sensor structure [14,15,16]. The phase and polarization of light waves are directly modulated by the applied external electric field based on the physical effects of the medium, thereby loading the electric field information to the light waves [17,18]. However, because the Kerr coefficient of the medium is small, the sensitivity of such a sensor is low [19]. Furthermore, as the independent optical components, such as the polarizer, quarter-wave plate, and prism, are required, the sensor structure is so complicated that it limits its practical application [20]. In addition, the measurement accuracy is affected by the alignment problem from the independent optical components.

To solve the aforementioned problem, in the 1980s, the integrated optical waveguide electric field sensor based on the Mach–Zehnder interferometer (MZI) was proposed for the first time and extensively researched [21,22,23]. Subsequently, the researchers mainly focused on how to improve the bandwidth, sensitivity, and stability of the sensor [24,25,26,27]. Recently, the sensors based on the bow-tie antenna with capacitive extended bars [28], multi-arm bow-tie antenna [29], and flexible bow-tie antenna [30] have been developed. The bandwidth of the sensor is optimized by a special antenna. In addition, the silicon–organic hybrid (SOH) technology [31] is used for integrated optical electric field sensors as well. Due to the slow light effect in the organic hybrid crystal optical waveguide (OWG), the light-matter interaction is enhanced [32,33]. Moreover, the patch-antenna embedded with a gap has been fabricated [34]. As the enhanced electric field in the narrow gap of the patch antenna directly modulates the light waves, the sensitivity is improved. The *x*-cut lithium niobate (LiNbO_3_, LN) integrated OWG sensor with an antenna has been designed to measure intensive pulsed electric fields [35,36]. However, these studies mainly concentrated on the influence of the antenna structure and substrate material on the bandwidth, sensitivity, and flatness of the sensor.

In the previous work, we have found that the dimension of the LN substrate and package have an influence on the resonant frequency of the sensor. The discovery has been verified by simulation and experiment, and the sensor bandwidth was optimized through simulation conclusions. In this paper, the influence of the LN substrate (length, width, thickness) and the package on the resonant frequency is analyzed. According to the simulation results, the parameters of the LN substrate, the relative permittivity of packaging material, and OWG are determined. An integrated OWG electric field sensor has been designed, fabricated, and experimentally investigated.

## 2. Sensor Structure and Simulation

### 2.1. Sensor Structure and Operation Principles

The schematic diagram of the sensor is shown in Figure 1. From Figure 1a, the sensor consisted of a tapered antenna, an OWG, a parallel modulation electrode, and an x-cut LN crystal. Figure 1b is the top view of the tapered antenna and the parallel modulation electrode. The OWG with width of 6 μm and depth of 4 μm was embedded into the LN crystal. SiO_2_ was used for a buffer layer over the LN crystal. Above the SiO_2_, the tapered antenna and parallel modulation electrode were fabricated by sputtering and electroplating technology. The narrow gap was used to amplify the induced electric field. Figure 1c is a magnified x-z plane cross-section view of the sensor. The point in the middle of the OWG was set as a probe to detect the electric field strength *E_in_* and resonance frequency *F_r_*.

The electro-optic (EO) coefficient tensor *γ*_33_ of the LN crystal is the largest. The dielectric constant in the z-axis of the LN crystal is 28. Therefore, when an electric field along the z-axis is applied to the x-cut LN crystal, a higher modulation efficiency will be produced [32,34].

When the wireless EM wave is captured by the tapered antenna, a standing wave current is generated on the surface of the tapered antenna. Due to the narrow gap in the middle of the modulation electrode, a displacement current is generated at the narrow gap. The displacement current induces a highly enhanced local electric field *E_in_* along the z-axis. When the induced electric field acts on the channel OWG, the refractive index of the channel OWG changes with the induced electric field, which causes the phase of the light in the channel OWG to be changed. As a result, the light in channel OWG is modulated by the EM field.

Under the highly enhanced local electric field *E_in_*, the refractive index difference Δ*n* can be obtained as
(1)$$\Delta n=\frac{1}{2}n_{eff}^{3}\gamma_{{}^{33}}\Gamma E_{in},$$
where *n_eff_* is the effective refractive index of the OWG, *γ_33_* is the EO coefficient, and *Γ* (<1) is the overlap factor of the electric field and optical field. The phase difference Δ*φ* of the light traveling in the two arms of the Mach–Zehnder interferometer (MZI) can be written as
(2)$$\Delta \varphi =\frac{2\pi }{\lambda }\Delta nL_{el}=\frac{\pi n_{eff}^{3}\gamma_{{}^{33}}\Gamma KE(t)L_{el}}{\lambda },$$
where *λ* is the wavelength of the input lights, *E*(*t*) is the incident electric field along the z-axis, *L_el_* is the length of the modulation electrode, and *K* is the electric field gain. When Δ*φ*(*E*) = π, the half-wave electric field *E*_π_ can be expressed as
(3)$$E_{\pi }=\frac{\lambda }{n_{eff}^{3}\gamma_{33}K\Gamma L_{el}}.$$

Based on the previous work [36], when π*E*(*t*)*/E_π_* is small and the bias phase *φ*_0_ = π/2, the output current *i_s_*(*t*) of the PD can be obtained as
(4)$$i_{s}(t)\approx \frac{1}{2}RP_{in}\alpha [1+k\cos\varphi_{0}-k\frac{\pi E(t)}{E_{\pi }}\sin\varphi_{0}],$$
where *P_in_* is the input optical power, *α* is the loss coefficient of the sensor element, *k* is the optical coefficient of the sensor element, *R* is the commutation coefficient of the light detection device. The relationship of the output voltage *V_out_, i_s_*(*t*) and *E*(*t*) can be obtained as
(5)$$V_{out}=\eta i_{s}(t)\propto E\left(t\right),$$
where *η* is the conversion coefficient between the voltage and the current. It can be seen from Equation (5) that the output voltage is proportional to the measured electric field. Therefore, the measured electric field can be obtained from the output voltage.

The mean square noise current of the PD can be expressed as
(6)$$<i_{n}^{2}>=<i_{t}^{2}>+<i_{d}^{2}>+<i_{sn}^{2}>+<i_{RIN}^{2}>,$$
where $<i_{t}^{2}>=4K_{b}TBR_{L}$ is the mean square thermal noise current caused by the random motion of electrons in the PD, $<i_{d}^{2}>=2e<i_{d}>B$ is the mean square shot noise current generated by the random distribution of photoelectrons in the PD, $<i_{sn}^{2}>=2e<i>B$ is the mean square dark noise current formed by hot carriers, $<i_{RIN}^{2}>=1/2<i{>}^{2}10^{RIN/10}$ is the mean square relative intensity noise current of the laser source, where *K_b_* is the Boltzmann constant, *T* is the absolute temperature, *B* is the noise bandwidth, *R_L_* is the load impedance, *e* is the unit electron charge, and *RNI* is the relative intensity noise of laser source.

From Equations (4) and (6), the signal to noise ratio (SNR) of the sensing system can be written as
(7)$$\frac{S}{N}=\frac{<i_{s}^{2}(t)>}{<i_{n}^{2}>}.$$

Therefore, when the SNR = 1, the minimum measurable electric field *E*_min_ of the sensor can be obtained as
(8)$$E_{\min}=\frac{2E_{\pi }\sqrt{\frac{4K_{b}TB}{R_{L}}+4eB<i_{d}>+R\alpha P_{in}(1+\cos\varphi_{0})eB+\frac{1}{8}{\left[R\alpha P_{in}(1+\cos\varphi_{0})\right]}^{2}10^{\frac{RIN}{10}}B}}{\pi R\alpha P_{in}\sin\varphi_{0}}.$$

### 2.2. Sensor Simulation Model

Figure 2 shows the diagram of the sensor simulation model established by ANSYS HFSS. A x-cut and y-propagating LN crystal with length *L_LN_* = 50 mm, width *W_LN_* = 6 mm, and thickness *T_LN_* = 0.5 mm was established as the substrate of the sensor. A LN OWG with width *W_wg_* = 6 μm and depth *D_wg_* = 4 μm was embedded in the LN crystal. In order to reduce the influence of the light absorption of the metal electrode on the sensor, a SiO_2_ buffer layer with a thickness of 4000 Å was added above the LN substrate. Above the SiO_2_ layer, there was a tapered antenna with arm length *L_a_* = 2.5 mm, bottom width *W_a_* = 1 mm, and thickness *T_a_* = 1 μm. The bottom of the tapered antenna was connected with the parallel modulation electrode with length *L_el_* = 5 mm and width *W_el_* = 20 μm. The width of the narrow gap was 10 μm in a parallel modulation electrode. The sensor was packaged by the polypropylene and plexiglass with relative permittivity of 2.15 and 3.4.

### 2.3. Simulation and Analysis

Consider a continuous wave (CW) polarized along the z-axis as an irradiation source. When the frequency of the incident EM wave is changed, the amplitude maintained remains constant. A point in the middle of the OWG is designed as a probe to detect the electric field strength *E_in_* and resonance frequency *F_r_*.

Figure 3 shows the resonant frequency of the sensor with the same tapered antenna but different LN substrate and package. For *T_LN_* = 0.5 mm and without package, the resonant frequency of the sensor was 6.84 GHz. For *T_LN_* = 0.5 mm and with package, the resonant frequency of the sensor was 6.39 GHz. For *T_LN_ =* 1 mm and without package, the resonant frequency of the sensor was 5.99 GHz. For *T_LN_ =* 1 mm and with package, the resonant frequency of the sensor was 5.58 GHz. From Figure 3, the largest resonant frequency was induced for the *T_LN_* = 0.5 mm and without package. For the sensor with package, the resonant frequencies were both reduced under *T_LN_* = 0.5 mm and *T_LN_* = 1 mm. When *T_LN_* was increased from 0.5 mm to 1 mm, the resonant frequencies of the sensor both packaged and without package were reduced.

In the next section, the relationships between the resonance frequency and the LN substrate and relative permittivity of the packaging material are analyzed. The simulation result is shown in Figure 4. From Figure 4a, when the width of the LN substrate was increased from 5.5 mm to 9 mm, the resonant frequency decreased from 6.841 GHz to 5.975 GHz. The resonant frequency was reduced by 12.7%. From Figure 4b, when the thickness was increased from 0.2 mm to 0.9 mm, the resonant frequency decreased from 9.173 GHz to 6.073 GHz. The resonant frequency was reduced by 33.4%. From Figure 4c, when the length of the LN substrate was increased from 40 mm to 58 mm, the resonant frequency fluctuated around 6.8 GHz, and the deviation was ±0.36 dB. From Figure 4d, when *ε_r_* increased from 1 to 9, the resonance frequency decreased from 6.765 GHz to 6.258 GHz. The resonance frequency was reduced by 7.49%.

Meanwhile, in order to reduce the transmission loss of the OWG MZI and improve the efficiency of the sensor, MZI was designed by Rsoft. As shown in Figure 5, The MZI was composed of an input straight waveguide, Y-branch waveguide, and cosine curved waveguide. The input straight waveguide with a length of 5 mm was designed to improve the stability of light transmission from the fiber to the OWG. To reduce the transmission loss, the ratio between the bend length *L*_2_ and the bend height *h*_2_ of the Y-branch waveguide needed to satisfy *L*_2_^2^/*h*_2_ > 250 [37,38]. Here, the bend height and length were designed as *h*_2_ = 12.5 μm, *L*_2_ = 1400 μm. It was considered that the total length of the MZI was 50 mm, the length *L*_1_ of the cosine bend waveguide was 18.6 mm. The relationship between the height and length of the cosine bending waveguide can be described as Equation (9). The length difference between the two arms of the MZI was designed as Δ*L* = 32 μm to ensure finding the best working point of the sensor [39]. From Equation (9), when Δ*L* = 32 μm, the height *h*_1_ of the cosine curved waveguide can be calculated as 464 μm.
(9)$$\Delta L=2\int_{0}^{L_{2}}\sqrt{1+(\frac{\pi^{2}h^{2}}{l^{2}}\sin^{2}(\frac{\pi x}{L^{{}_{2}}}))}dx-2L_{2}.$$

The simulation results of the OWG MZI transmission characteristics are shown in Figure 5. Figure 5a is the simulation diagram of Y-branch waveguide. From Figure 5b, the total transmission loss of the designed MZI was 0.1188 dB. The designed parameters of the integrated OWG electric field sensor are shown in Table 1.

## 3. Sensor Fabrication and Experiment

### 3.1. Fabrication and Packaging

According to parameters in Table 1, the integrated OWG electric field sensor with LN substrate thickness of 1 mm was fabricated and packaged. The fabrication processes are shown in Figure 6. First, an x-cut y-propagating LN crystal was chosen as the substrate. The OWG MZI was fabricated on the LN crystal by Lithography and annealing proton-exchange (APE) technology. In order to ensure that the proton exchange only occurred in the waveguide region, ~70 nm Cr barrier layer was fabricated by electroplating technology. Then photoresist was covered on the Cr barrier layer. The pattern of the mask was developed by photoresist. The proton exchange window was fabricated by etching. The proton exchange window is shown in Figure 6e. Benzoic acid with a small amount of lithium was used as a proton source. The LN crystal was exchanged in a high-temperature resistant glass tube at ~300 °C for ~4.5 h. Then the waveguide was annealed in an annealing furnace at ~350 °C for ~5 h. The cross-sectional view of LN OWG is shown in Figure 6g. Second, the SiO_2_ buffer layer was fabricated by radio frequency magnetic sputtering (RFMS) on the OWG. Third, the Au tapered antenna and parallel modulation electrode were fabricated on the SiO_2_ buffer layer by sputtering technology and electroplating technology.

Polypropylene with a relative permittivity of 2.3 was used as the package material to ensure the sensor had a higher resonance frequency. Figure 7 is the photograph of the packaged integrated OWG electric field sensor. The size of the packaged sensor was 78 × 18 × 7.5 mm^3^.

### 3.2. Frequency Response Measurement

Figure 8 shows the schematic of the sensor frequency response measurement experimental setup. The microwave generator (ROHDE & SCHWARZ^®^SMA100B) with a frequency range of 8 kHz to 40 GHz was used as the microwave source. By using two power amplifiers (RFLIGHT 10 kHz~6 GHz and RFLIGHT 6 GHz~18 GHz), the microwave signal was amplified and transmitted to the coupler. The microwave signals output from the coupler were transmitted to the horn antenna with bandwidth from 800 MHz to 18 GHz and the power meter by a constant proportion. The sensor received the microwave signal emitted by the horn antenna. A standard EO field strength meter was placed close to the sensor to monitor the electric field strength of the radiated microwave signal. The optical host1 consisted of a PD (KG-PD-20 G), a micro integrable tunable laser assembly (ITLA), a bias micro-control circuit (MCU), and an optical coupler (OC). The linear polarized light from the optical host1 was transmitted to the sensor through the polarization-maintaining fiber (PMF). Then the linear polarized light traveling through the OWG was modulated by the microwave signal. The output modulated optical signal was transmitted to the PD through the single mode fiber (SMF) and converted into electrical signal. The detected electric field signal was extracted by the spectrum analyzer (ROHDE & SCHWARZ^®^FSW43) with a frequency range of 2 kHz to 43.5 GHz.

As shown in Figure 9a, the horn antenna was used to radiate the microwave signal. Meanwhile, it can be seen in Figure 9b, the electric field of the radiated microwave signal was detected by the sensor, and simultaneously, the electric field strength was measured by the field strength meter. Consider the distance between the horn antenna with the sensor was *d* > 2D^2^/λ, the radiation power density *P_s_* can be expressed as
(10)$$P_{s}=\frac{GP}{4\pi d^{2}},$$
where D is the caliber of the horn antenna, λ is the wavelength of the radiation microwave signal, *G* is the gain of the horn antenna, and *P* is the input power of the microwave signal. The electric field strength near the sensor can be obtained.
(11)$$E=\sqrt{P_{s}Z}=\frac{1}{2d}\sqrt{\frac{GP}{\pi }Z},$$
where *Z* is the spatial wave impedance. According to Equation (11), *G* = 16 dBi, *P* = 13 dBm, *Z* = 377 Ω, *d* = 5 m, the field strength of the microwave signal was calculated as 8 V/m.

The simulation and experiment results of the frequency responses are shown in Figure 10. The measured resonance frequency of 7.5 GHz was approximately equal to the simulated resonance frequency of 7.12 GHz. In addition, the variation tendencies of the experimental results and the simulation results were nearly consistent.

### 3.3. Time-Domain Response Measurement

Moreover, the time-domain response of the sensor was measured under the nanosecond electromagnetic pulse (NEMP). The experimental measurement setup is shown in Figure 11. The electric field sensor was fixed in the TEM cell (ROHDE & SCHWARZ TEMZ5233) with a septum height of d = 20 cm, and a bandwidth of DC-420 MHz. The TEM cell was connected with the nanosecond impulse generator (LSE-545CB) with a rise time t_r_ ≈ 5 ns. When output voltage U was provided by a nanosecond impulse generator, according to E_d_ = U/d, an intensive nanosecond pulsed electric field of 5U was generated in the TEM cell and acted on the optical signal was transmitted in OWG. The modulated optical signal was converted to an electrical signal by the PD and then obtained by an oscilloscope (TEKTRONIX MSO54) with a bandwidth of 1 GHz and a real-time sampling rate of 6.25 GS/s. At the same time, the pulse waveform generated by the impulse generator was transmitted to the oscilloscope after attenuation to compare with the detected signal.

The comparison diagram of the waveform detected by the sensor and the input waveform is shown in Figure 12. The yellow and the blue represent the input voltage waveform and the electric field waveform detected by the sensor, respectively. Figure 12a shows the time-domain response of the sensor under the electric field of 13 kV/m. The rise time and pulse width of the detected waveform were 5 ns and 24.4 ns. Meanwhile, the rise time and pulse width of the input waveform were 5.1 ns and 24.3 ns. The relative errors of the rise time and pulse widths were 2% and 0.4%. Figure 12b is the time-domain response of the sensor under the electric field of 25 kV/m. The rise time and pulse bandwidth of the detected waveform were 6.1 ns and 22.7 ns. The rise time and pulse width of the input waveform were 6.2 ns and 22.8 ns. The relative errors of the rise time and pulse widths were 1.6% and 0.4%. The output waveform of the sensor and the input waveform almost coincided, but there was a delay of about 110 ns. This is mainly caused by the difference in the transmission speed and propagation path length of optical signals in cables and optical fibers.

### 3.4. Dynamic Linear Range

The input/output characteristics of the sensor are shown in Figure 13. It can be seen that the electric field provided by NEMP had a better linear relationship with the output of the sensor in the range of 1 kV/m to 25 kV/m. The linear fitting correlation coefficient was 0.9992. Under a signal to noise ratio (SNR) of not more than 3 dB, the minimum measurable electric field of the sensor was 1.29 kV/m. In addition, the electric field gain K = E_in_/E(t) = 6.2 was calculated by ANSYS HFSS. From Equations (3) and (8), E_π_ = 336.56 kV/m and E_min_ = 1.146 kV/m could be obtained. The theoretical minimum measurable electric field was less than the actual measurement. It is considered to be caused by the incomplete consideration of noise source. The thermal noise, the shot noise, the dark noise, and the relative intensity noise were considered as all noise sources in the theoretical calculation, while the sensing system also included 1/f noise, generation recombination (g-r) noise, and temperature noise, etc., due to the maximum electric field 25 kV/m which could be provided by the laboratory. The maximum measurable electric field E_max_ = 100.97 kV/m (E_max_ = 0.3E_π_) of the sensor was calculated. The measurable linear range of the sensor was 20 dB.

## 4. Conclusions

The simulation results showed that the thickness of the LN substrate had the greatest impact on the resonance frequency. The results of frequency response measurement suggested that the measured resonance frequency of 7.5 GHz was approximately equal to the simulated resonance frequency of 7.12 GHz, which verified the correctness of the simulation. Therefore, the resonant frequency and bandwidth of the sensor can be optimized by designing the dimension of the LN substrate. Moreover, the results of the time-domain response measurement showed that the sensor can be used for measuring the intense nanosecond electromagnetic pulse electric field. In addition, the sensor had a good linear relationship between input and output, and the linear measurement range was 20 dB (1.29 kV/m~200.97 kV/m). The significance of sensor simulation and measurement proposed in this paper is to provide effective guidance for designing OWG electric field sensors.

## Figures and Tables

**Figure 1 sensors-21-03672-f001:**
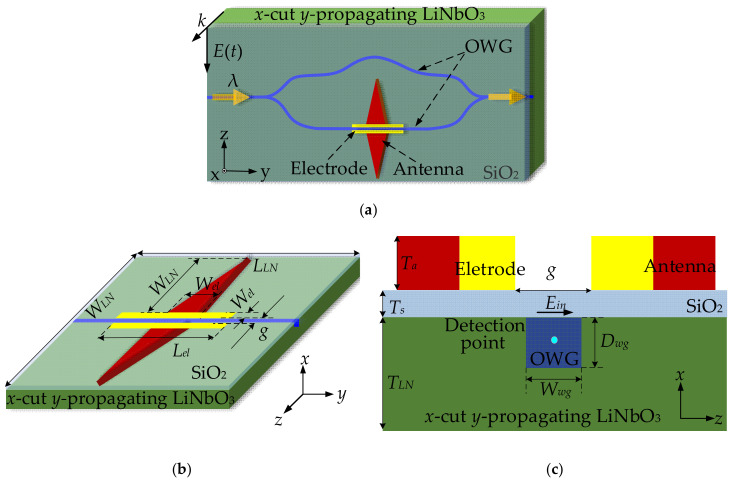
Schematic of the integrated optical electric field sensor: (**a**) sensor structure; (**b**) antenna and electrode; (**c**) x-z plane cross section.

**Figure 2 sensors-21-03672-f002:**
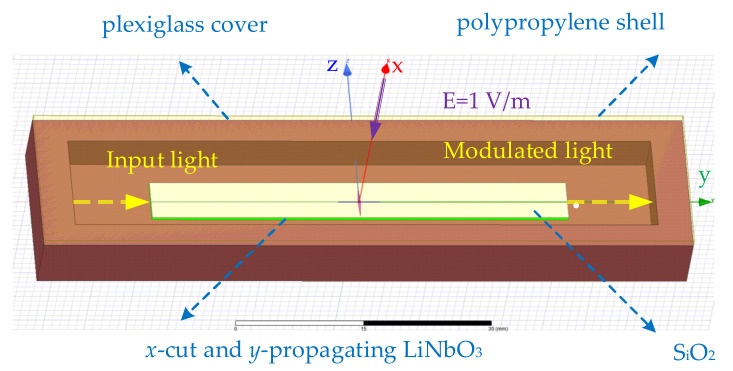
Diagram of the sensor simulation model.

**Figure 3 sensors-21-03672-f003:**
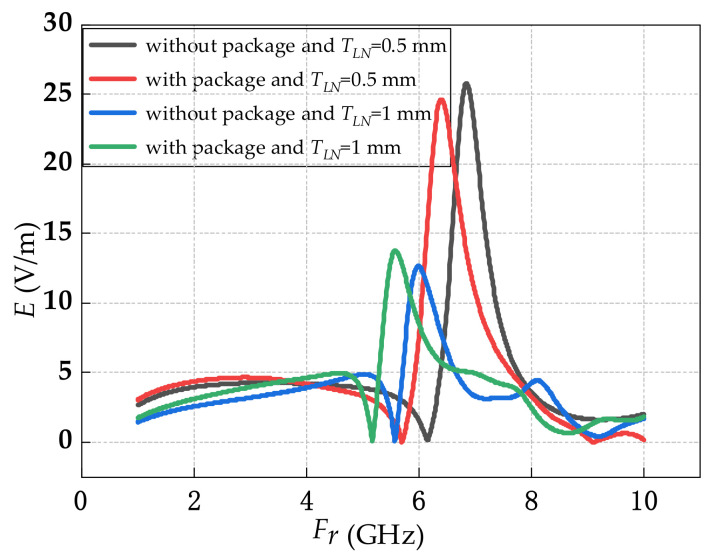
Different structures resonant frequency.

**Figure 4 sensors-21-03672-f004:**
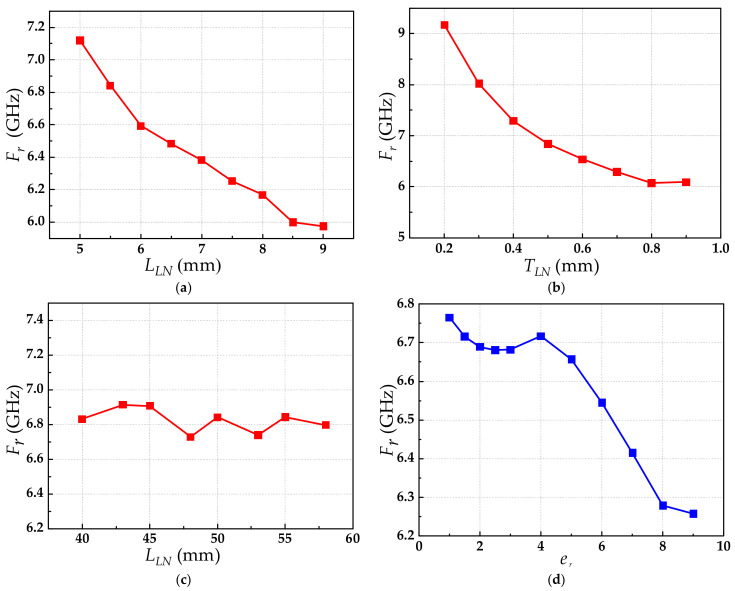
The relationships between the resonance frequency *F_r_* and the LN substrate and the relative permittivity of the packaging material: (**a**) *F_r_* versus *W_LN_*; (**b**) *F_r_* versus *T_LN_*; (**c**) *F_r_* versus *L_LN_*; (**d**) *F_r_* versus *ε_r_*.

**Figure 5 sensors-21-03672-f005:**
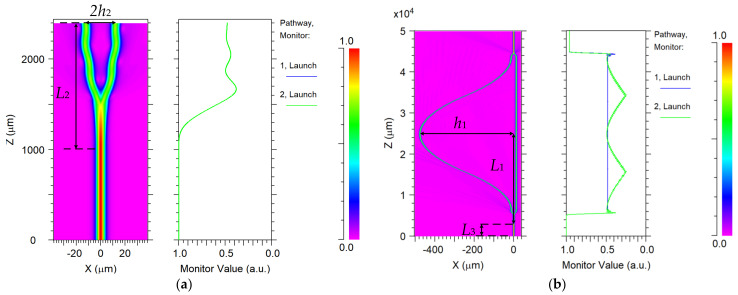
Transmission characteristics of OWG: (**a**) Y-branch OWG; (**b**) integrated OWG.

**Figure 6 sensors-21-03672-f006:**
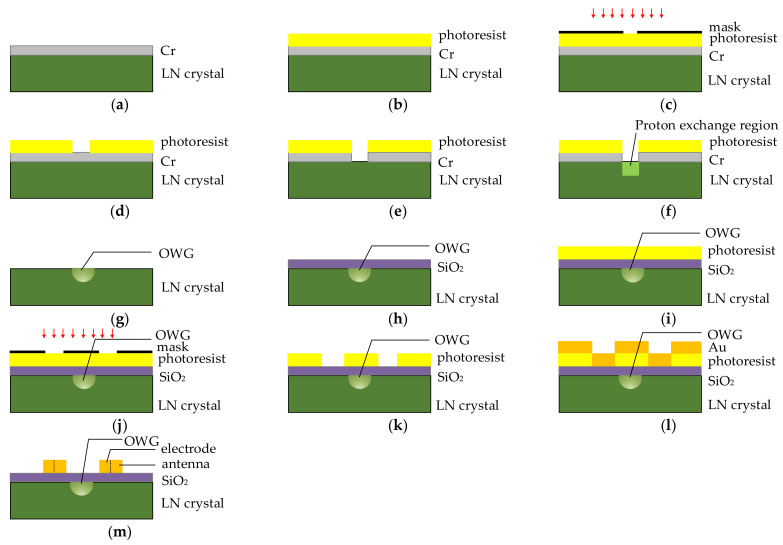
Fabricate process: (**a**) Cr-plated barrier; (**b**) coat with photoresist; (**c**) align and expose; (**d**) develop; (**e**) etch; (**f**) proton exchange; (**g**) finished OWG; (**h**) Sputter SiO_2_ buffer layer; (**i**) coat with photoresist again; (**j**) align and expose again; (**k**) develop again; (**l**) plate antenna and electrode; (**m**) ultrasonic stripping.

**Figure 7 sensors-21-03672-f007:**
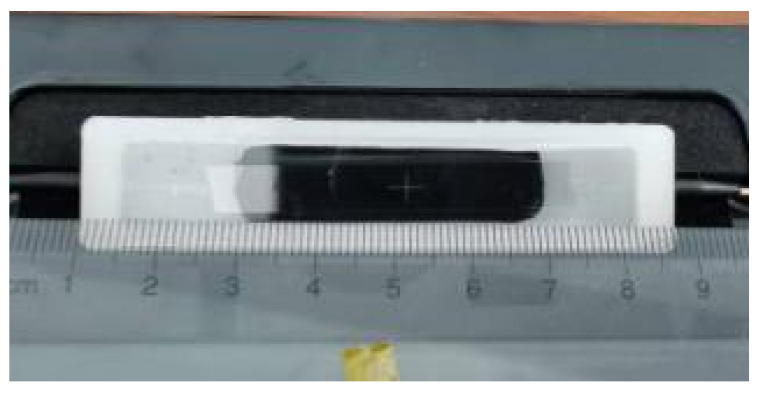
Photograph of the packaged sensor.

**Figure 8 sensors-21-03672-f008:**
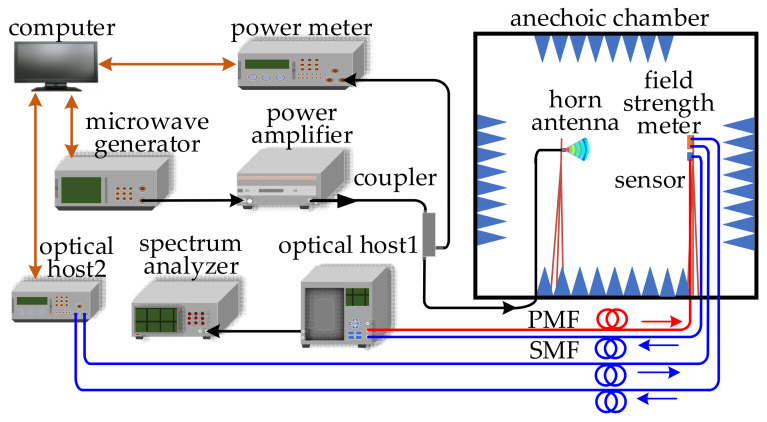
Schematic of sensor frequency response measurement setup.

**Figure 9 sensors-21-03672-f009:**
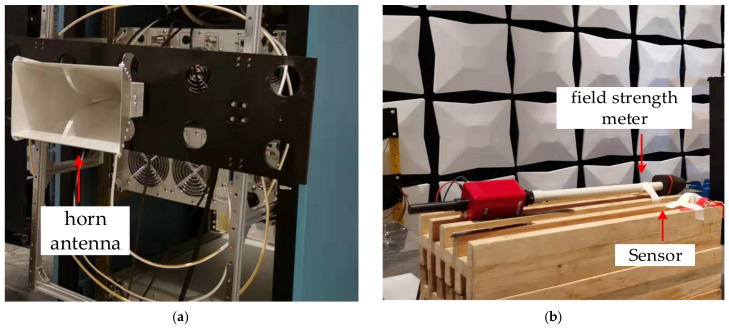
Photograph of sensor frequency response measurement setup: (**a**) radiation area; (**b**) receiving area.

**Figure 10 sensors-21-03672-f010:**
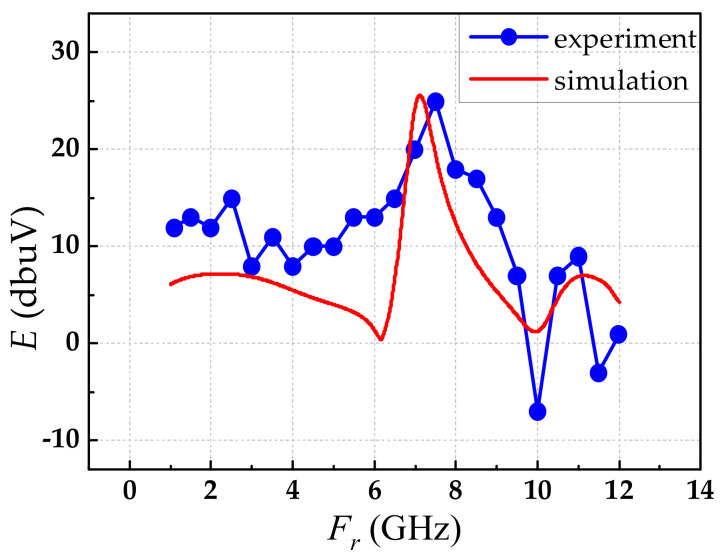
The frequency response of the experiment and simulation.

**Figure 11 sensors-21-03672-f011:**
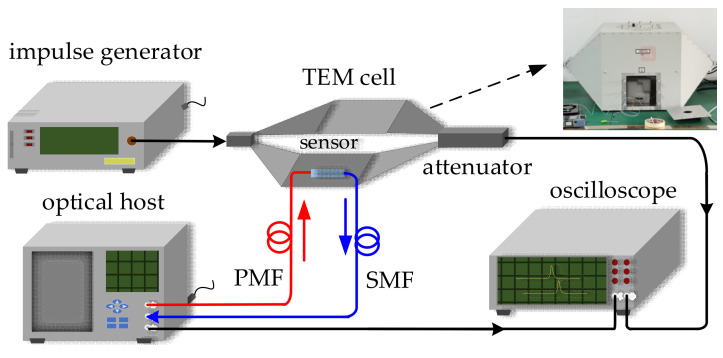
Schematic of time-domain response measurement setup.

**Figure 12 sensors-21-03672-f012:**
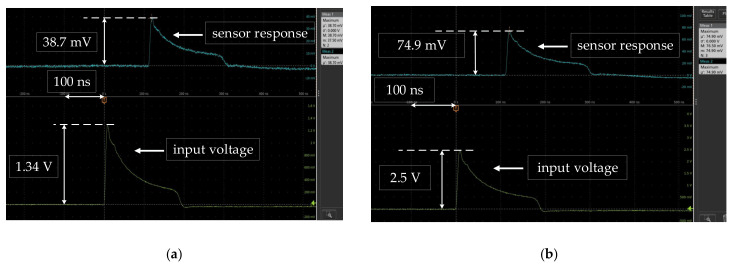
Time-domain response (100 ns/div): (**a**) E = 13 kV/m; (**b**) E = 25 kV/m.

**Figure 13 sensors-21-03672-f013:**
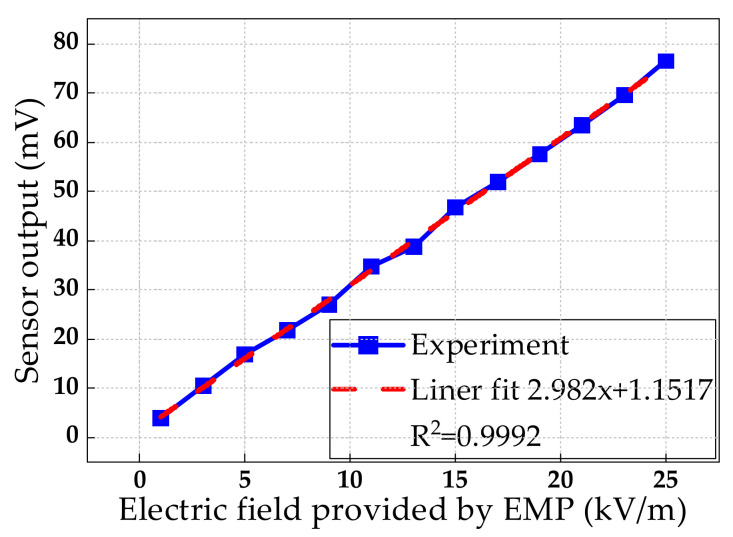
Sensor input/output characteristics.

**Table 1 sensors-21-03672-t001:** The parameters of the OWG electric field sensor.

Structure Name	Length		Width		Thickness	
	Symbol	Value (mm)	Symbol	Value (mm)	Symbol	Value (mm)
LN substrate	*L_LN_*	50	*W_LN_*	5	*T_LN_*	1
Electrode	*L_el_*	5	*W_el_*	2 × 10^−2^	*T_el_*	10^−3^
Tapered antenna	*L_a_*	2	*W_a_*	1	*T_a_*	10^−3^
Silica buffer layer	*L_s_*	50	*W_s_*	5	*T_s_*	4 × 10^−7^
package	*L_pa_*	78	*W_pa_*	18	*H_pa_*	7.5
LN OWG	*L_wg_*	50	*W_wg_*	6 × 10^−3^	*S_wg_*	4 × 10^−3^

## Data Availability

No database required.

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
