# Peer review of "Design and Analysis of Broadband LiNbO3 Optical Waveguide Electric Field Sensor with Tapered Antenna"

_sensors, 2021, doi:10.3390/s21113672_

Round 1

Reviewer 1 Report

This paper presented the influences of the LN substrate and the packaging material on the resonance frequency of the integrated OWG electric field sensor. The simulation results show that the increase the LN substrate size and relative permittivity, the decrease the resonance frequency. The experimental result has good consistency with the simulated result. I suggested that this paper can be accepted. However, some minor errors and extensive English correction are needed before publication. For example, in line 246, The blue and the yellow (Fig.11) represent the input waveform and the waveform detected by the sensor, respectively. No yellow color was observed and blue color was indicated the response waveform. Please, check it.  

Reviewer 2 Report

The article "Design and analysis of broadband LiNbO3 optical waveguide electric field sensor with tapered antenna" present the designed, fabricated, and experimentally investigation of an integrated optical waveguide electric field sensor. The proposed work is good, but in this reviewer’s opinion, the paper needs improvements:

1- In the Abstract avoid using the software name of the private industries (ANSYS HFSS). Use "full-wave electromagnetic simulation software". After, in the text of the article, you use ANSYS HFSS.  

2- Insert the models of equipment used in sensor frequency response measurement setup and in time domain response measurement setup.

3- What is the power of the laser source?

4- Why use PMF fiber in the input of the sensor and SMF fiber in the output sensor?

5- If possible insert photos of measurement setups.

Reviewer 3 Report

The paper under consideration provides a numerical and experimental study of the Electric field sensor based on LN waveguide. I am in favour to accept this paper for publication. However, the paper has written in a semi-scientific language and full of English grammatical errors. I suggest the authors take help from a native English speaker or some services from the Journal to rectify the errors. Moreover, I have the following few suggestions which should also be addressed.

  • The structure of the Introduction section is not appropriate. I suggest the authors divide the big paragraph into small paragraphs and extend the bibliography. At the moment, the Introduction section is weak and unacceptable.
  • I am not sure why the author has emphasized numerical simulation results in the Abstract and Conclusion section. Although the author has presented sound experimental results. It is better to highlight Experimental results more than numerical in both sections. The paper is attractive as the experimental study is provided.
  • The reference section should be extended and make a fair comparison with the previously reported Electric field sensors. And explain why your sensor device is better than others.
  • Please provide the RSoft Beam Prop simulation parameters such as boundary conditions, etc.
  • In figure 9, the author mentioned that “The experimental result have good consistency with the simulated result.” But I am not sure if this is the case, as the two graphs are not coinciding. Please explain.
  • The paper needs a thorough English correction. For instance “The experimental result have good consistency with the simulated result.” -à the experimental result has ……..

Round 2

Reviewer 3 Report

Accept in its current form.